# ASSANet: An Anisotropic Separable Set Abstraction for Efficient Point Cloud Representation Learning

Guocheng Qian      Hasan Abed Al Kader Hammoud      Guohao Li      Ali Thabet

**Bernard Ghanem**
King Abdullah University of Science and Technology (KAUST)
{guocheng.qian, hasanabedalkader.hammoud, bernard.ghanem}@kaust.edu.sa
https://github.com/guochengqian/ASSANet

## Abstract

Access to 3D point cloud representations has been widely facilitated by LiDAR sensors embedded in various mobile devices. This has led to an emerging need for fast and accurate point cloud processing techniques. In this paper, we revisit and dive deeper into PointNet++, one of the most influential yet under-explored networks, and develop faster and more accurate variants of the model. We first present a novel Separable Set Abstraction (SA) module that disentangles the vanilla SA module used in PointNet++ into two separate learning stages: (1) learning channel correlation and (2) learning spatial correlation. The Separable SA module is significantly faster than the vanilla version, yet it achieves comparable performance. We then introduce a new Anisotropic Reduction function into our Separable SA module and propose an Anisotropic Separable SA (ASSA) module that substantially increases the network's accuracy. We later replace the vanilla SA modules in PointNet++ with the proposed ASSA module, and denote the modified network as ASSANet. Extensive experiments on point cloud classification, semantic segmentation, and part segmentation show that ASSANet outperforms PointNet++ and other methods, achieving much higher accuracy and faster speeds. In particular, ASSANet outperforms PointNet++ by $7.4$ mIoU on S3DIS Area 5, while maintaining $1.6\times$ faster inference speed on a single NVIDIA 2080Ti GPU. Our scaled ASSANet variant achieves 66.8 mIoU and outperforms KPConv, while being more than $54\times$ faster.

## 1 Introduction

Among the various 3D object representations, point clouds have been surging in popularity, becoming one of the most fundamental 3D representations. This popularity stems from the increased availability of 3D sensors, like LiDAR, which produce point clouds as their raw output. The growing presence of point cloud data has been accompanied by the development of many 3D deep learning methods [28, 41, 19, 38, 22]. Even though these methods achieve impressive performance, they are generally computationally expensive (Figure 1). With the integration of LiDAR sensors into hardware-limited devices, such as mobile devices and AR headsets, interest in efficient models for point cloud processing has grown significantly. Given the limited computational power of mobile devices and embedded systems, the design of mobile-friendly point cloud-based algorithms should not only focus on providing good accuracy, but also on maintaining high computational efficiency.

When processing point cloud data, one can always opt to convert the data into representations that are more accessible to deep learning frameworks. Popular options are multi-view methods [34, 5, 42] and voxel-based methods [6, 47]. Converting to these representations generally requires additional

35th Conference on Neural Information Processing Systems (NeurIPS 2021).

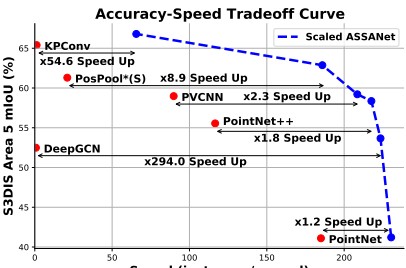

Figure 1: **Tradeoffs between accuracy (mIoU on S3DIS Area-5) and inference speed (instances/second)**. Speed is reported as the mean value of 200 runs on a single GTX 2080Ti GPU. The proposed ASSANet scaled with different widths and depths shown in **- - -** outperform the state-of-the-art methods in ● with better accuracies and faster speeds. Refer to Section 5.2 for details.

computation and memory, and can lead to geometric information loss [23]. It is therefore more desirable to operate directly on point clouds. To that extent, we are currently witnessing a surge in point-based methods [27, 28, 41, 19, 38, 22]. The first of such methods was introduced by Qi *et al.* through the seminal PointNet [27] architecture. PointNet operates directly on point clouds, without the need for an intermediate representation. Despite its efficiency, PointNet merely learns per-point features individually and discards local information, which restrains its performance. As a variant of PointNet, PointNet++ [28] presents a novel *Set Abstraction* module that sub-samples the point cloud, groups the neighborhood, extracts local information via a set of multi-layer perceptrons (MLPs), and then aggregates the local information by a reduction layer (*i.e.* pooling). Figure 1 shows how PointNet++ outperforms the pioneering PointNet [27] by a large margin. PointNet++ also obtains better accuracy than the graph-based method DeepGCN [19], and does so with a $100\times$ speed gain. PointNet++ provided a good balance between accuracy and efficiency, and was therefore widely utilized in various tasks like normal estimation [8], segmentation [26, 14], and object detection [32]. After PointNet++, graph-based [33, 39, 41, 19], pseudo-grid based [37, 20, 25, 38] and adaptive weight-based [40, 21, 7, 44], became the state-of-the-art in point cloud tasks. As shown in Figure 1, nearly all of these methods improve performance at the cost of speed. In this work, we focus on designing point cloud networks that are both fast and accurate. Inspired by its success, both in terms of the accuracy-speed balance and its wide adoption, we take a deep dive into PointNet++. We conduct extensive analysis of its architectural design (Section 3.1) and latency decomposition (Figure 2). Interestingly, we demonstrate that both its efficiency and accuracy can be improved sharply by minimal modifications to the architecture. These modifications lead to a new architecture design that is faster and more accurate than currently available point methods (shown in **- - -** in Figure 1).

**Contributions.** **(1)** We demonstrate that the MLPs performed on the neighborhood features in the Set Abstraction (SA) module of PointNet++ reduce the inference speed. We introduce a new **separable SA** module that processes on point features directly allowing for a significant improvement in inference speed. **(2)** We discover that all operations for processing neighbors in the SA module are isotropic which limits the performance (accuracy wise). We present a novel **Anisotropic Reduction** layer that treats each neighbor differently. We then insert Anisotropic Reduction into our Separable SA and propose the Anisotropic Separable Set Abstraction (ASSA) module that greatly increases accuracy. **(3)** We present ASSANet by replace the vanilla SA in PointNet++ with the proposed ASSA. ASSANet shows a much higher accuracy and a faster speed compared to PointNet++ and previous methods on various tasks (point cloud classification, semantic segmentation, and part segmentation). We further study two regimes for up-scaling ASSANet. As shown in Figure 1, *our scaled ASSANet outperforms the previous state-of-the-art with a much faster inference speed*. In particular, scaled ASSANet achieves better accuracy than the graph-based method DeepGCN [19] with an increase in speed of $294\times$, the pseudo grid-based method KPConv [38] ($54\times$ faster), the adaptive weight-based method PosPool*(S) [22] ($9\times$ faster), and the efficient 3D method PVCNN [23] ($2\times$ faster).

## 2 Related Work

**Projection-based methods.** Due to the unstructured nature of point clouds, convolutional neural networks (CNNs) that tend to work impressively well on grid stuctured data (*e.g.* images, texts and videos) fail to apply directly on point clouds. One common solution for processing point clouds is to project them into collections of images (views) [34, 5, 42] or 3D voxels [6, 47, 36]. Common CNN backbones (using 2D or 3D convolutions) can be subsequently utilized to perform these intermediate representations. Although projection-based methods allow for utilizing the well studied convolutional neural networks to point-cloud applications, they are computationally expensive as they are associated

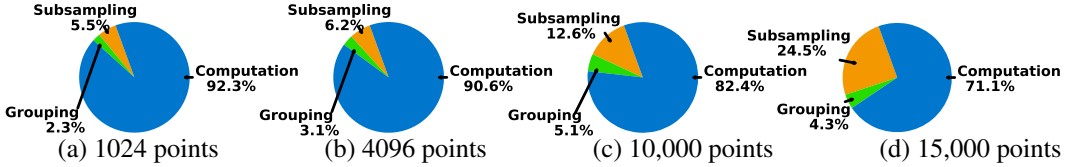

Figure 2: **Latency Decomposition of PointNet++.** We show the inference run time decomposition of PointNet++ under different numbers of points as input on one NVIDIA GTX2080Ti GPU.

with additional cost of constructing intermediate representations. Moreover, the projection of point clouds causes loss of important geometric information [23].

**Point-based methods.** Pioneering work explored the possibility of processing point clouds directly. Qi *et al*. proposed PointNet [27] that leverages point-wise MLPs to extract per point features individually. To better encode locality, Qi *et al*. further presented Set Abstraction (SA) to aggregate features from the points' neighborhood, and a hierarchical architecture named PointNet++ [28] that learns multilevel representations and reduces the required computations. After PointNet++, numerous point-based methods considering neighborhood information were proposed. Graph-based methods [33, 15, 41, 39, 19, 18, 29] represent point clouds as graphs and process point clouds with graph neural networks. Pseudo grid-based methods project neighborhood features onto different forms of pseudo grids such as tangent planes [37], grid cells [11, 46, 20, 25, 38] and spherical grid points [49] which allow convolving with regular kernel weights like CNNs. Adaptive weight-based methods perform weighted neighborhood aggregation by considering the relative positions of the points [40, 21, 7, 22] or point density [44]. These methods rely either on designing sophisticated and customized modules, which usually require expensive parameter tuning for different applications [38, 20, 24], or on performing expensive graph kernels [41, 19] that achieve better performance than PointNet and PointNet++ at the expense of computational complexity.

**Efficient Neural Networks.** Efficient neural networks is a class of architectures that target mobile and embedded systems applications. These networks are usually designed to provide a balance between accuracy and efficiency (*e.g.* latency, FLOPs, memory, and power). MobileNet [9] utilizes depth-wise separable convolutions to reduce the required FLOPs and latency of a regular CNN for image processing. Depth-wise separable convolutions disentangle convolutions into learning channel correlations using point-wise convolutions and learning spatial correlations using depth-wise convolutions. Other efficient neural networks usually leverage either depth-wise separable convolutions with better designed architectures to improve performance [31, 4, 48] or study new efficient operations to replace the regular convolutions [24, 43]. In 3D, efficient neural networks include ShellNet [49], PVCNN [23], Grid-GCN [45], RandLA-Net [10], SegGCN [17] and LPNs [16]. ShellNet [49] and SegGCN [17] speed up the pseudo grid-based methods by aggregating neighborhood features through efficient 1D convolutions or fuzzy spherical convolutions on the predefined pseudo grids like shells. PVCNN [23] and Grid-GCN [45] reduce the time spent in querying a neighborhood by combining voxelization in point-based methods. RandLA-Net [10] reduces the subsampling complexity by leveraging random sampling and further improves the speed by operating on a large-scale point cloud directly without chunking. LPN [16] improves the speed of convolving neighborhood features by a simple group-wise matrix multiplication. Nevertheless, all efficient methods mentioned above require performing convolutions on neighborhood features, which we deem through extensive experiments as unnecessary. Therefore, our algorithm achieves much faster speeds compared to these methods (ref to Section 4). It is also worthwhile to mention that our method can be made even faster with the voxelization trick in PVCNN and Grid-GCN to further reduce the latency of neighborhood querying. We leave that as future work.

## 3 Methodology

### 3.1 Preliminary: PointNet++

PointNet++ [28] improves PointNet [27] by providing two main contributions: (1) developing a U-Net [30] like architecture to process a set of points, which are sampled in a metric space in a hierarchical fashion. This mechanism captures multi-scale features and reduces the required

computation. (2) Developing a Set Abstraction (SA) module to process and abstract the locality from the local neighbors to a new set of points with fewer elements. The SA module is used as the basic building block to be stacked to form the backbone of PointNet++.

**Analysis of the Set Abstraction Module.** As illustrated in Figure 3a, the vanilla SA module proposed in PointNet++ consists of two parts: point subsampling and feature aggregation, . The subsampling layer takes a point cloud $X = \{P, F\}$ as an input and leverages iterative farthest-point sampling to acquire $X'$, a subset of $X$. $P$ and $F$ denote the coordinates and features, respectively. The feature aggregation block is built for learning locality from local neighbors and is composed of a grouping layer, an MLP block, and a reduction layer. The grouping layer obtains the neighborhood composed of $K$ neighbors for each point in $X'$ using the ball query, with $X$ as the support set. The resulting point neighborhood is denoted as $\mathcal{N}(X')$, which contains $K$ repeated features. The MLP block consists of $L$ layers of MLPs, and each MLP is followed by a Batch Normalization (BN [12]) layer and a ReLU activation. By default, PointNet++ sets $L = 3$. The number of feature aggregation blocks inside one SA module, referred to as depth $D$ in this paper, is set to $D = 2$. The reduction layer (*a.k.a*, pooling) aggregates the neighborhood information by a reduction function, *e.g.* mean, max, or sum. The feature aggregation is formulated as shown in Equation (1):

$$\mathbf{f}_i^{l+1} = \mathcal{R}\left(\{\text{MLPs}((\mathbf{p}_j - \mathbf{p}_i)||\mathbf{f}_j^l)|j \in \mathcal{N}(i)\}\right), \tag{1}$$

where $\mathcal{R}$ is the reduction function across the neighborhood dimension, which is used for aggregating the neighborhood information. $\mathbf{p}_i, \mathbf{f}_i^l, \mathcal{N}(i)$ and $||$ denote the coordinates, the features in the $l^{th}$ layer of the network, the neighborhood of the $i^{th}$ point, and the concatenation operator across the channel dimension, respectively. The main issues with the vanilla SA module are: **(1)** the computational cost is unnecessarily high. *MLPs are unnecessarily performed on the neighborhood features*, which causes a considerable amount of latency in PointNet++. One straightforward remedy is to use MLPs to learn a feature embedding on the point features directly instead of doing so on the neighborhood features. This reduces the FLOPs of each MLP by a factor of $K$. **(2)** *All operations on neighbors are unnecessarily isotropic.* In other words, the MLPs and the reduction layer treat all local neighbors equally. This severely limits the representation capability of the network.

**Latency Decomposition.** Figure 2 shows the latency decomposition of PointNet++ [28] with different numbers of points as input. Here, the latency, which is the overall run time for the inference stage, was measured using a single Nvidia GeForce RTX 2080Ti GPU and one Intel(R) Xeon(R) CPU E5-2687W v4 @ 3.00GHz. We note here that latency is measured on the same hardware setting throughout this work. The latency of PointNet++ can be decomposed into three main contributing factors: (1) point subsampling, (2) grouping, (3) actual computations. The actual computations of PointNet++ mainly come from processing neighborhood features by MLPs shown in Equation 1. Note that we consider the time spent on data access implicitly in each part. Point clouds with four different input sizes were studied: 1024, 4096, 10,000, and 15,000. The first two input sizes are commonly encountered in classification tasks [2], and the last two are usually input sizes for patch-based segmentation [28, 38] and LiDAR-based object detection [32]. Clearly, computations contribute to the majority of latency (over 70 %). This suggests that *the computational complexity could be the major speed bottleneck for networks involving PointNet++.*

### 3.2 Anisotropic Separable Set Abstraction (ASSA)

In this section, we gradually introduce the modified vanilla SA modules. Initially, we focus on speeding up vanilla SA. This is achieved through proposing two modules, namely, PreConv SA module and Separable SA module. Later, we focus our attention on improving the accuracy by proposing an Anisotropic SA module.

**PreConv Set Abstraction module.** Vanilla SA repeatedly performs shared MLPs on point neighborhood features. To solve this issue, we modify the feature aggregation layer in vanilla SA, and propose PreConv SA to performs all MLPs on point features directly (not on the $k$ local neighbors) before the grouping layer. The PreConv SA is shown in **Appendix** Figure S1, and its feature aggregation is formulated as follows:

$$\mathbf{f}_i' = \text{MLPs}\left(\mathbf{f}_i^l\right), \mathbf{f}_i^{l+1} = \mathcal{R}\left(\{\mathbf{f}_j' \mid j \in \mathcal{N}(i)\}\right) \tag{2}$$

PreConv SA reduces the required FLOPs by $K$ times. PreConv SA speeds up PointNet++ by $\sim 55\%$ (15,000 points), as shown in Section 5.1. Additionally, PreConv SA is equivalent to vanilla SA in the case where the $(\mathbf{p}_j - \mathbf{p}_i)$ term is not included in Equation (1). Proof is available in the **Appendix**.

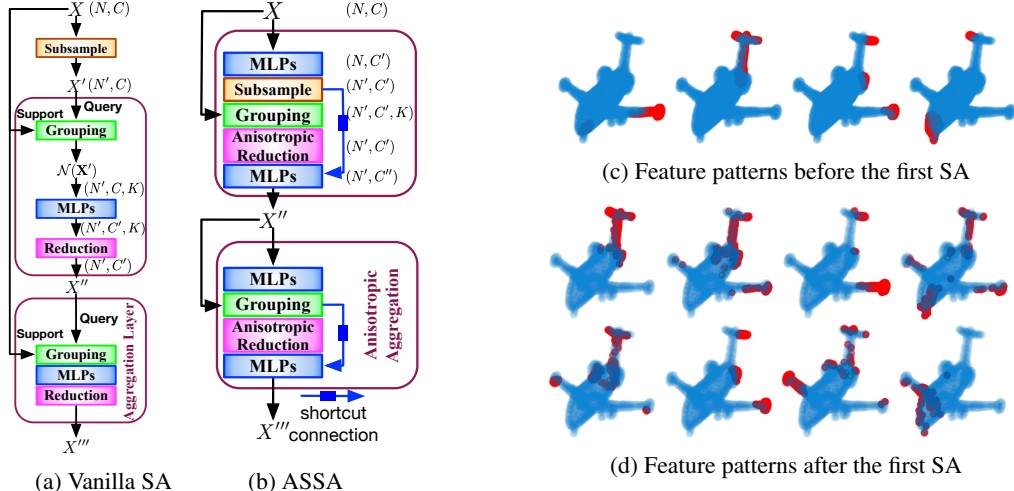

(a) Vanilla SA    (b) ASSA

(c) Feature patterns before the first SA

(d) Feature patterns after the first SA

Figure 3: **Comparison of proposed Anisotropical Separable Set Abstraction (ASSA) module and the Vanilla SA module.** (a) Vanilla SA [28] applies MLPs on neighbor features. (b) The proposed ASSA module separates the MLPs before the grouping layer and after the reduction layer. Therefore the MLPs are applied directly on the point features not on the neighbor features. ASSA also replaces the reduction layer in vanilla SA with a new Anisotropic Reduction layer. $X, N, C, K$ are the input point cloud, the number of points, the number of input features, and the number of neighbors. The shortcut layer in blue line is the residual connection with a linear mapping. (c) and (d) Show the point cloud feature patterns (activations) before and after the first ASSA module. The proposed ASSA module helps capture better geometric relationships. Refer to Section 5.1 for details.

**Separable Set Abstraction module.** Next, we present Separable Set Abstraction (refer to **Appendix** Figure S1), which is more accurate than PreConv SA, yet requires the same latency. The aggregation layer in Separable Set Abstraction is formulated by:

$$\mathbf{f}_i^{res} = \text{MLPs}\left(\mathbf{f}_i^l\right), \mathbf{f}_i^{l+1} = \mathbf{f}_i^{res} + \text{MLPs}\left(\mathcal{R}\left(\{\mathbf{f}_j^{res} \mid j \in \mathcal{N}(i)\}\right)\right) \tag{3}$$

The main idea of Separable SA is borrowed from depth-wise separable convolutions [9], where the regular convolution is split into one point-wise convolution (MLPs), one depth-wise convolution (channel shared convolution), and then another point-wise convolution. Separable SA evenly separates the MLPs before the grouping layer and after the reduction layer and further adds a residual connection between the outputs of the two parts of the MLPs. The main reasons why the Separable Set Abstraction module is better than PreConv are: **(1)** after reduction MLPs further process the aggregated neighborhood information; **(2)** The residual connection not only stabilizes training, but also provides better feature embedding by fusing the aggregated local information with the point information. Another minor change from PreConv SA to Separable SA is that we query the neighborhood using the subsampled point cloud $X'$ as the support set to further reduce the computational complexity of the second aggregation block.

**Anisotropic Separable Set Abstraction module.** PreConv SA and Separable SA cut down computational complexity at the expense of accuracy, *e.g.* Separable SA leads to a reduction of 3 mIoU on S3DIS Area 5 compared to PointNet++ (Section 5.1). There are two reasons for the drop in accuracy. First, the geometric information is not well encoded in the current variants of the SA module. The geometric information can be represented by any relative information (edge information) between the neighbor and the center, *e.g.* the relative position $(\mathbf{p}_j - \mathbf{p}_i)$ in PointNet++. The experiments in Section 5.1 show that geometric information is essential for point feature embedding. Second, the reduction layer is an isotropic operation that treats each neighbor the same and thus leads to a sub-optimal representation. Recall in a depth-wise separable convolution, the depth-wise convolution uses different weights to summarize features from a $3 \times 3$ receptive field. However, simply introducing the depth-wise convolution kernel to point neighborhood aggregation does not work, as: (1) the neighbors are not necessarily ordered for the sake of efficiency; (2) the convolution kernel is shared by all points and neighbors and leads to poor neighborhood aggregation where the local geometric varies. We propose an efficient geometric-aware *Anisotropic Reduction* layer to effectively

aggregate the point neighborhood information. The term "Anisotropic" indicates that our reduction layer considers each neighbor differently. We insert Anisotropic Reduction into the separable SA module and present our final variant of the SA module, the Anisotropic Separable Set Abstraction (ASSA) module, and show it in Figure 3b. The feature aggregation of ASSA is formulated as:

$$\mathbf{f}_i^{res} = \text{MLPs}(\mathbf{f}_i^l)$$

$$\mathbf{f}_i^{l+1} = \text{LN}\left(\mathbf{f}_i^{res}\right) + \text{MLPs}\left(\mathcal{R}\left(\left\{\left(\frac{\Delta x_{ij}\mathbf{f}_j^{res}||\Delta y_{ij}\mathbf{f}_j^{res}||\Delta z_{ij}\mathbf{f}_j^{res}}{r}\right)|j \in \mathcal{N}(i)\right\}\right)\right) \quad (4)$$

$\Delta x_{ij} = x_j - x_i$, $\Delta y_{ij}$ and $\Delta z_{ij}$ are the relative positions between the neighbor $j$ and the center $i$ in the $x, y, z$ dimension, respectively. The relative positions are used as scaling weights for aggregating the features across the neighborhood dimension, and they are normalized by the radius of the ball query $r$. The neighborhood features are scaled by the three corresponding relative positions individually. The three scaled neighborhood features are then concatenated together and passed into the reduction layer. To reduce the computational complexity caused by the concatenation of the three scaled features, we set the last MLP before reduction as a bottleneck layer. This layer reduces the number of channels by a factor of 3. The output of the reduction layer is then processed by another MLP block and is added to the output before reduction. Due to channel mismatch, the output of before grouping MLPs is mapped by a linear layer LN (*a.k.a* the shortcut layer) before the addition. We highlight that our Anisotropic Reduction does not rely on any heuristic grouping (as done in PosPool [22]), and we make full use of the information from the neighborhood features. The pseudo code for ASSA in PyTorch-like style is available in the **Appendix**.

It is worth noting that all MLPs in ASSA are processed on the point features directly, not on the neighborhood, which greatly reduces the computations compared to Equation (1). In particular, for one aggregation block with $L = 3$ MLPs, ASSA roughly reduces the FLOPs consumed in vanilla SA by: $\frac{C \times C \times N \times K \times L}{C \times C \times N \times L + C \times N \times K} \approx K$ times. Typically, $K$ is around 32. All of our SA variants are permutation invariant, which favors 3D deep learning on point clouds. More details of the ASSA module and its comparison with previous modules are provided in the **Appendix**.

### 3.3 ASSANet

We now replace the vanilla SA module in PointNet++ [28] with our proposed ASSA module. All other parts are kept the same as PointNet++, including the number of SA modules (4), the number of aggregation blocks in SA ($D = 2$), the layers of MLPs in an aggregation block ($L = 3$), the channel sizes, the neighborhood querying configurations (ball query algorithm with maximum neighborhood size $K$ and radius $r$) and the subsampling configurations (farthest point sampling). The modified architecture of PointNet++ is referred to as ASSANet. Section 4 shows that ASSANet can achieve much higher accuracies compared to PointNet and PointNet++ and is faster on various vision tasks.

### 3.4 Scaling ASSANet

Since the ASSANet is much faster than both PointNet++ [28] and the state-the-of-art networks, we now present two ways to up-scale ASSANet to improve its accuracy: width scaling and depth scaling. We show the performance of each scaling regime in the ablation study presented in Section 5.2.

**Width Scaling Regime.** In width scaling regime, we modified the channel size of ASSANet. ASSANet is built upon PointNet++ [28], which uses hand-crafted channel sizes for each convolution layer. To make the scaling more programmable and user-friendly for the scaled ASSANet, the output of each feature aggregation block inside one ASSA module is set to have the same channel size, and is then concatenated as the output of the module. After this modification, we can easily study the effect of width scaling on the accuracy and the speed, by simply changing the initial channel size $C$.

**Depth Scaling Regime.** The second way to scale is to increase the depth of the network, which can be achieved by changing the number of aggregation blocks $D$ stacked in each ASSA module. $D$ is set to 2 by default in ASSANet. We can decrease $D$ to 1 to make ASSANet faster or increase $D$ to improve its accuracy. Among all width or depth scaled versions of ASSANet, we emphasize **ASSANet (L)**, a large ASSANet network with $C = 128$ and $D = 3$. In most of the experiments, we compare ASSANet and ASSANet (L) with the state-of-the-art.

## 4  Experiments

We studied the accuracy and speed of ASSANet and ASSANet (L) on S3DIS semantic segmentation [1], ShapeNet part segmentation [3], and ModelNet40 point cloud classification [2]. To enable a fair comparison, the same data processing and evaluation protocols adopted by the state-of-the-art method PosPool [22] were used in our experiments.

### 4.1  3D Scene Segmentation

**Setups.** We conducted extensive experiments on the Stanford large-scale 3D Indoor Spaces (S3DIS) dataset [1]. Following [20, 23, 22], we trained all our models on Area 1, 2, 3, 4, and 6 and tested them on Area 5. We optimized all

| Methods | mIOU % | Inference Speed instances/second |
|---|---|---|
| PointNet [27] | 41.1 | 185.0 |
| DeepGCN [19] | 52.5 | 0.8 |
| PointCNN [20] | 57.3 | 124.1 |
| Grid-GCN [45] | 57.8 | 123.5 |
| PVCNN [23] | 59.0 | 89.8 |
| PosPool*(S) [22] | 61.3 | 21.0 |
| SegGCN [17] | 63.6 | 29.3 |
| KPConv [38] | 65.4 | 1.2 (24.2) |
| PosPool* [22] | 66.7 | 8.3 |
| PointNet++ [28] | 55.6 | 116.6 |
| ASSANet | 63.0 (+7.4) | 188.6 (1.6×) |
| ASSANet (L) | 66.8 (+11.2) | 65.6 |

Table 1: **S3DIS scores (mIoU) on Area-5.** ASSANet outperforms PointNet++ and other methods with much higher accuracy and faster speed. ASSANet (L) performs better than the state-of-the-art KPConv [38] and PosPool* [22] while being over 7.9× faster.

of our networks using SGD with weight decay 0.001, momentum 0.98 and initial learning rate (LR) 0.02. We trained the models for 600 epochs and used an exponential LR decay. At each inference time, a single RTX 2080Ti GPU was used to measure the speed for each method using a batch size of 16; each item in the batch has $15,000$ points ($16 \times 15,000$). If the batch size was too large to feed into the GPU, we lowered the batch size. *Note that we focus on the speed since FLOPs and the model parameter size are not indicative of the actual latency [24, 23].* The inference speed is calculated as the number of instances evaluated in one second (ins./sec.). The average speed over 200 runs is reported. Other methods were measured in a similar manner. Note that KPConv [38] has to compute the pseudo kernels for each point cloud during data preprocessing. For a fair comparison, we show the speed of calculating the pseudo kernels on the fly. We also include the speed of KPConv with preprocessed pseudo kernels in () in the table.

**Comparison with state-of-the-art.** Table 1 compares the proposed ASSANet and ASSANet (L) with PointNet++ [28] and the state-of-the-art on S3DIS. *ASSANet outperforms PointNet++ by 7 mIoU and is $1.6\times$ faster.* ASSANet also achieves much better accuracy than the two efficient point cloud processing algorithms PVCNN [23] and Grid-GCN [45], while also being over $1.5\times$ faster. *ASSANet (L) achieves state-of-the-art performance with a mIoU of 66.8% on S3DIS, with very high speed.* ASSANet (L) is $294\times$ faster than the graph-based method DeepGCN [19], $54.6\times$ faster than the state-of-the-art pesudo grid-based method KPConv [38], $7.9\times$ faster than the state-of-the-art adaptive weight-based method PosPool* [22], and $2.2\times$ faster than the best-performing efficient method SegGCN [17]. Note that PosPool* refers to PosPool with sinusoidal position weight, and that PosPool* (S) denotes the small model.

### 4.2  3D Object Classification

**Setup.** As a common practice, we benchmark ASSANet on the ModelNet40 [2] object classification dataset. We adopted a similar training setting as that of S3DIS except that we used LR 0.001 and a cosine LR decay. At the inference time, a single RTX 2080Ti GPU was used to measure the speed for the classification task using $16 \times 10,000$ points as input.

**Comparison with state-of-the-art.** Table 2 compares ASSANet and ASSANet (L) with the state-of-the-art. ASSANet outperforms PointNet++ by 1.7 units in overall accuracy and is $2.1\times$ faster than PointNet++. ASSANet (L) achieves on par accuracy as the state-of-the-art methods KPConv [38] and PosPool* [22] while being $5.0\times$ and $4.4\times$ faster, respectively.

### 4.3  3D Part Segmentation

**Data.** ShapeNetPart is a commonly used benchmark for 3D part segmentation. The networks were optimized using Adam [13] with momentum 0.9. The other training parameters were the same as

| Methods | OA % | Inference Speed instances/second |
|---|---|---|
| PointNet [27] | 89.2 | 483.8 |
| SpiderCNN [46] | 90.5 | < 275.7 |
| PointCNN [20] | 92.5 | 183.4 |
| PosPool*(S) [22] | 92.6 | 48.8 |
| DGCNN [41] | 92.9 | 11.6 |
| KPConv [38] | 92.9 | (30.1) |
| Grid-GCN [45] | 93.1 | 172.0 |
| PosPool* [22] | 93.2 | 27.6 |
| PointNet++ [28] | 90.7 | 275.7 |
| ASSANet | 92.4 (+1.7) | 586.4 (2.1×) |
| ASSANet (L) | 92.9 (+2.2) | 153.2 |

Table 2: **Comparison of our ASSANet and ASSANet (L) with other methods on ModelNet40 point cloud classification.** ASSANet outperforms PointNet++ with 1.7 higher overall accuracy (OA) than PointNet++ and is 2.1 times faster. ASSANet (L) achieves on par accuracy with the state-of-the-art while maintaining a high speed.

| Methods | mIoU % | Inference Speed instances/second |
|---|---|---|
| PointNet [27] | 83.7 | 1883.5 |
| PosPool* (S) [22] | 85.1 | 107.7 |
| DGCNN [41] | 85.2 | 151.4 |
| LPN [16] | 85.7 | 190.6 |
| PosPool* [22] | 85.8 | 58.0 |
| PointCNN [20] | 86.1 | 626.4 |
| RS-CNN [21] | 86.2 | <350.4 |
| KPConv [38] | 86.2 | (56.3) |
| PointNet++ [28] | 85.1 | 350.4 |
| ASSANet | 85.4 (+0.3) | 782.5 (2.2×) |
| ASSANet (L) | 86.1 (+1.0) | 438.5 (1.3×) |

Table 3: **Comparison of the part-averaged IoU (mIoU) of our ASSANet and AS-SANet (L) with other methods on ShapeNetPart part segmentation.** Both of ASSANet and ASSANet (L) outperform PointNet++ with a higher speed. ASSANet (L) achieves a comparable accuracy as the state-of-the-art while being much faster.

ModelNet40 experiments. The speed of each method was measured with an input of $16 \times 2048$ points. We report the part-averaged IoU (mIoU) as the evaluation metric for accuracy.

**Comparison with state-of-the-art.** Table 3 shows that ASSANet again outperforms PointNet++ with a sharp increase ($2.2\times$) in speed on the ShapeNetPart part segmentation dataset. ASSANet (L) also achieves 1 unit higher mIoU than PoinetNet++ with a $1.3\times$ faster speed. Additionally, ASSANet (L) attains on-par accuracy, $86.1\%$ mIoU, with the state-of-the-art and is much faster. For example, ASSANet (L) is nearly $7.8\times$ faster than KPConv [38].

# 5 Ablation Study

An ablation study was conducted on S3DIS [1] Area-5. We show the effectiveness of the proposed SA variants and the effect of the two scaling regimes on ASSANet.

## 5.1 Ablation on Proposed SA variants

Table 4 shows the speed and the accuracy of the proposed PreConv Set Abstraction (SA) module, the Separable SA module, and the Anisotropic Separable SA (ASSA) module compared to the vanilla SA. All of our proposed SA modules lead to a sharp increase (over $1.6\times$) in inference speed. The proposed Separable SA module can boost the

| Aggregation | mIoU % | Speed ins./sec. |
|---|---|---|
| Vanilla SA | 55.6 | 116.6 |
| PreConv SA | 48.7 | 180.9 |
| Separable SA (SSA) | 52.4 | 180.0 |
| SSA + Relative Position | 58.5 | 184.0 |
| SSA + Attentive Pooling[10] | 59.0 | 142.0 |
| SSA + PosPool[22] | 62.0 | 168.4 |
| **Anisotropic Separable SA** | **63.0** | **188.6** |

Table 4: **Ablation study of the proposed SA variants.** All proposed SA variants achieve a faster speed than the vanilla SA. Our ASSA further improves the accuracy of the Separable SA module and outperforms other methods, while also being faster.

accuracy of PreConv by 3.7 mIoU, which verifies the effectiveness of separable MLPs and residual connections. Comparing the ASSA module with the Separable SA module, one can clearly see the importance of encoding the geometric information and the effect of the anisotropic operation to achieving higher accuracy. Additionally, we provide a comparison of our proposed Anisotropic Reduction with the Attentive Pooling used in RandLA-Net [10] and the PosPool proposed in [22]. Our method clearly outperforms both of these methods in terms of accuracy and inference speed. We also test simple addition of the relative positions $\Delta_x + \Delta_y + \Delta_z$ as the weights of the reduction layer, denoted as Relative Position, the obtained performance is worse than the proposed Anisotropic Reduction. To further show the benefits of the proposed ASSA module, we visualize the **feature patterns** before and after the ASSA module in Figure 3. ASSA module helps capture better geometric relationships among points constituting the point cloud (for example, in the second and fourth

examples in the first row of Figure 3d, one can see that the ASSA module allows the network to learn relationships between the tail of the plane and its wings). **Appendix** provides a more detailed overview and a further set of examples of feature patterns visualization.

## 5.2 Ablation on Scaling Regimes

We now study the effects of ablating the width and depth of a network on its accuracy and inference speed. The initial channel size of the network is referred to as width (denoted by $C$), whereas the number of aggregation layers inside a single SA module is referred to as depth (denoted by $D$).

**Width scaling.** Figure 4 (left) shows the effect of the width scaling regime. When the width of the network is small, increasing the width leads to a significant improvement in accuracy. For example, simply increasing the width $C$ from 3 to 8 sharply improves the accuracy from 41.21 mIoU to 53.95 with a negligible drop in speed. However, when the network is wide enough ($C \geq 128$), increasing the width further only leads to a marginal improvement in accuracy, yet reduces the speed noticeably.

**Depth scaling.** Figure 4 (right) shows the effect of the depth scaling regime. We study the depth scaling with $C = 128$, which is the sweet point of width scaling. When the network is shallow, with a depth of $D \leq 3$, increasing the depth leads to an obvious increase in accuracy. However, depth scaling rapidly saturates as the depth increases. Depth scaling leads to a linear reduction in speed.

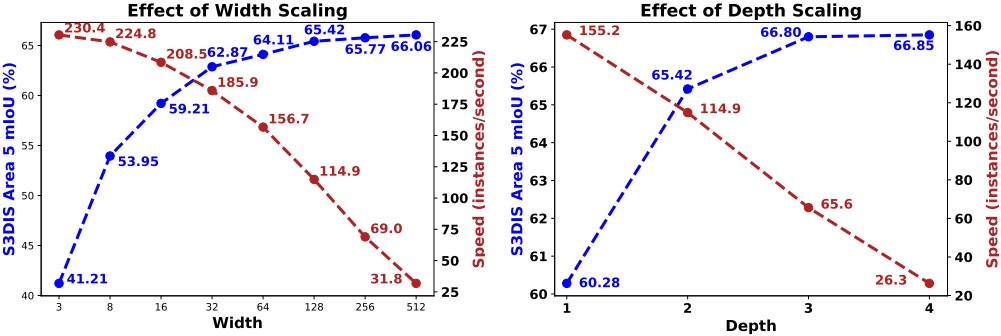

Figure 4: **Effect of Width (left) and Depth Scaling (right).** Increasing either the width or the depth leads to an improvement in accuracy and drop in inference speed.

## 6 Conclusion

In this paper, we dove deeper into the architecture of PointNet++. We noticed that PointNet++ suffers from a computational burden attributed to the MLPs that process the neighborhood features in the set abstraction (SA) module. We also found out that the accuracy of PointNet++ is limited by the isotropic nature of its operations. To solve these issues, we proposed a PreConv SA module, a Separable SA module, and finally an Anisotropic Separable SA (ASSA) module that aim to reduce the computational cost and improve the accuracy. We then replaced the vanilla SA module in PointNet++ with our ASSA module and proposed a fast and accurate architecture, namely, ASSANet. Extensive experiments were conducted to verify the presented claims and showed that ASSANet achieves largely improved accuracy and much faster speed on various point cloud tasks, such as classification, semantic segmentation, and part segmentation. We also studied up-scaling ASSANet. The scaled ASSANet set new state-of-the-art on various tasks with faster speeds. For future work, one could leverage both random sampling [10] and voxelization tricks [23, 45] to further improve the inference speed. Alternatively, one could consider studying compound scaling, like that in EfficientNet [35].

**Acknowledgement.** The authors appreciate the anonymous NeurIPS reviewers for their constructive feedback (including the revised title, the feature pattern visualization, and the additional experiments). This work was supported by the KAUST Office of Sponsored Research (OSR) through the Visual Computing Center (VCC) funding.

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
