# ASSANet: An Anisotropic Separable Set Abstraction for Efficient Point Cloud Representation Learning –Supplementary Material–

**Guocheng Qian**     **Hasan Abed Al Kader Hammoud**     **Guohao Li**     **Ali Thabet**

**Bernard Ghanem**
King Abdullah University of Science and Technology (KAUST)
{guocheng.qian, hasanabedalkader.hammoud, bernard.ghanem}@kaust.edu.sa
https://github.com/guochengqian/ASSANet

In this supplementary material, we provide a detailed analysis of the proposed Set Abstraction (SA) variants.

## A   More about SA Variants

### A.1   Equivalence between PreConv SA and vanilla SA

Our proposed PreConv SA is formulated in Equation 1.

$$
\begin{aligned}
\mathbf{f}'_i &= \mathrm{MLPs}\left(\mathbf{f}^l_i\right) \\
\mathbf{f}^{l+1}_i &= \mathcal{R}\left(\left\{\mathbf{f}'_j | j \in \mathcal{N}(i)\right\}\right),
\end{aligned}
\tag{1}
$$

Here, we show that the proposed PreConv SA is equivalent to the vanilla SA when edge information is not used. In this case, the vanilla SA is formulated as follows:

$$
\mathbf{f}^{l+1}_i = \mathcal{R}\left(\left\{\mathrm{MLPs}\left(\mathbf{f}^l_j\right) | j \in \mathcal{N}(i)\right\}\right),
\tag{2}
$$

Since (1) the neighborhood querying function $\mathcal{N}(i)$ is independent on the point features, and (2) the MLPs is a **shared** point-wise function processed on each neighbor features, Equation 1 is equivalent to Equation 2.

### A.2   Pseudocode of ASSA

We provide the pseudocode of the proposed ASSA module in Algorithm 1.

35th Conference on Neural Information Processing Systems (NeurIPS 2021), Sydney, Australia.

**Algorithm 1** Pseudocode of ASSA in a PyTorch-like style.

```
# f: input features with shape: (B,C,N)
# p_q, p_s: (x,y,z) position for query and support
# shortcut: nn.Linear(ceil(C/3), C)

# conduct MLPs on input point features
f = ReLU(MLP(f) + f) # (B,ceil(C/3),N)

# query the neighborhood features f_N (B,ceil(C/3),N,K)
# and the normalized relative position d_p (B,3,N,K)
f_N, d_p = Group(p_q, p_s, f)

# repeat f_N to size (B,3,ceil(C/3),N,K)
# f_N is then element-wise weighted by d_p
f_N = f_N.expand(3, dim=1) * d_p
f_N = f_N.view(B, C, N, K)

# Reduction layer aggregates the neighborhood information
# outputs f_aggr: (B,ceil(C/3)*3,N)
f_aggr = Reduction(f_N)

# MLPs on f_aggr and obtain f_out, output dimensions C
f_out = ReLU(MLP(f_aggr) + shortcut(f))
# return f_out with shape (B,C,N)
```

### A.3  Difference of ASSA with previous work

Our ASSA module can be viewed as a separable anisotropic graph convolution for point cloud learning. Inspired by the depthwise separable convolution used in MobileNet [1], we extend this separable idea to graph convolution. We only perform MLPs on point features directly to learn the channel correlation, and leverages the anisotropic reduction to aggregate the spatial correlation. Such anisotropic reduction was studied in adaptive weight-based point cloud learning methods like ParamConv [5], RS-CNN [2], *etc*. They used expensive computations to learn adaptive weights from the relative position or the local density. The recent paper by Liu [3] proposed PosPool, which simply leveraged the relative position vector **p** as the adaptive weights and outperformed previous methods. However, their method requires to divide the channels into three parts and scale the each part by $x, y, z$, respectively. This leads to performance degradation for two major reasons: first, the scaling is based on the heuristic grouping, and second, each scaling only sees one part of the feature instead of the whole information. On the contrary, our method does not conduct any grouping. The proposed Anisotropc Reduction scales the whole feature by three times, and then concatenate them together. We have shown the superiority of our Anisotropic Reduction and the ASSA module in terms of accuracy and speed in main paper Section 5.1.

### A.4  Illustration of PreConv SA and Separable SA

### A.5  Latency analysis of ASSA compared with vanilla SA

We show the latency decomposition of ASSA module only compared to vanilla SA module only. We show the cases using 4096 points as input and 15,000 points as input. In both case, our proposed ASSA module reduces the time consumed in computation part by $4\times$.

## B  Feature Patterns

Following PointNet++ [4], we also visualize what has been learned by the first ASSA module. We create a voxel grid in space and aggregate local point sets that activate certain neurons the most in grid cells (highest 100 examples are used). Grid cells with high votes are kept and converted back to 3D point clouds, which represents the pattern that neuron recognizes. We use objects from ModelNet40 as examples. Figure S3 - S7 show the feature patterns before and after the first ASSA module of airplane, car, chair, table, lamp, and person. These figures show that the proposed ASSA module helps capture better geometric relationships.

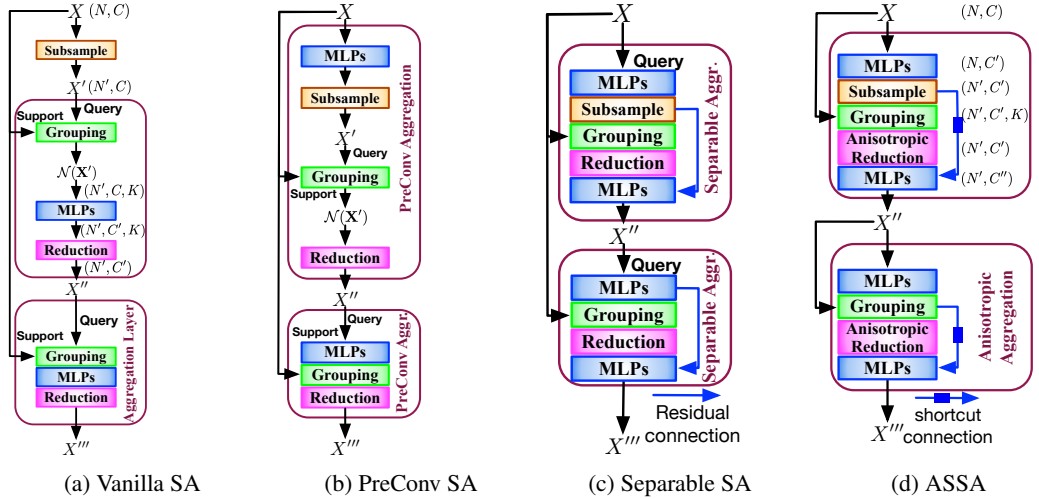

| (a) Vanilla SA | (b) PreConv SA | (c) Separable SA | (d) ASSA |
|---|---|---|---|

Figure S1: **Comparison of proposed variants of the Set Abstraction (SA) module and the Vanilla SA module.** (a) Vanilla SA [4] applies MLPs on neighbor features. (b) The proposed PreConv SA applies MLPs on the point features directly. (c) Our Separable SA separates the MLPs to also process on the aggregated features from a local neighbors. (d) Our final ASSA module replaces the reduction layer in Separable SA with a new Anisotropic Reduction layer. $X, N, C, K$ are the input point cloud, the number of points, the number of input features, the number of neighbors. The shortcut layer in blue line is the residual connection with a linear mapping. The shortcut layer is the residual connection with a linear mapping.

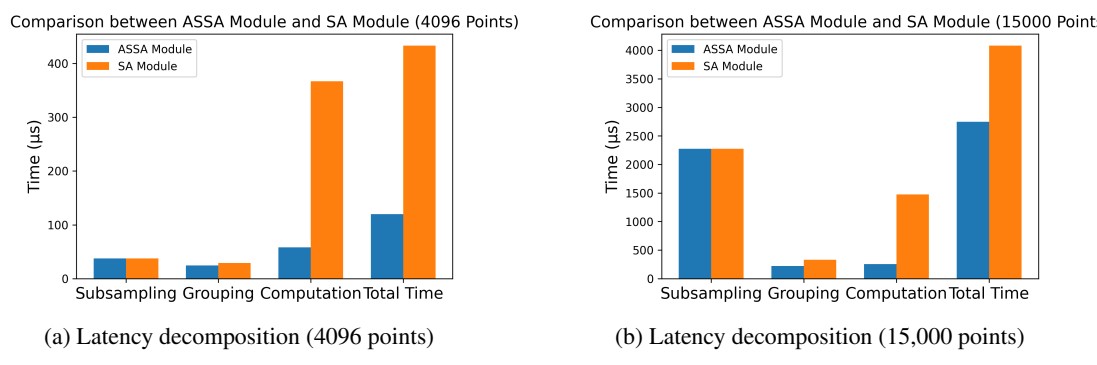

| (a) Latency decomposition (4096 points) | (b) Latency decomposition (15,000 points) |
|---|---|

Figure S2: **Latency decomposition of proposed ASSA module compared with vanilla SA module**.

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

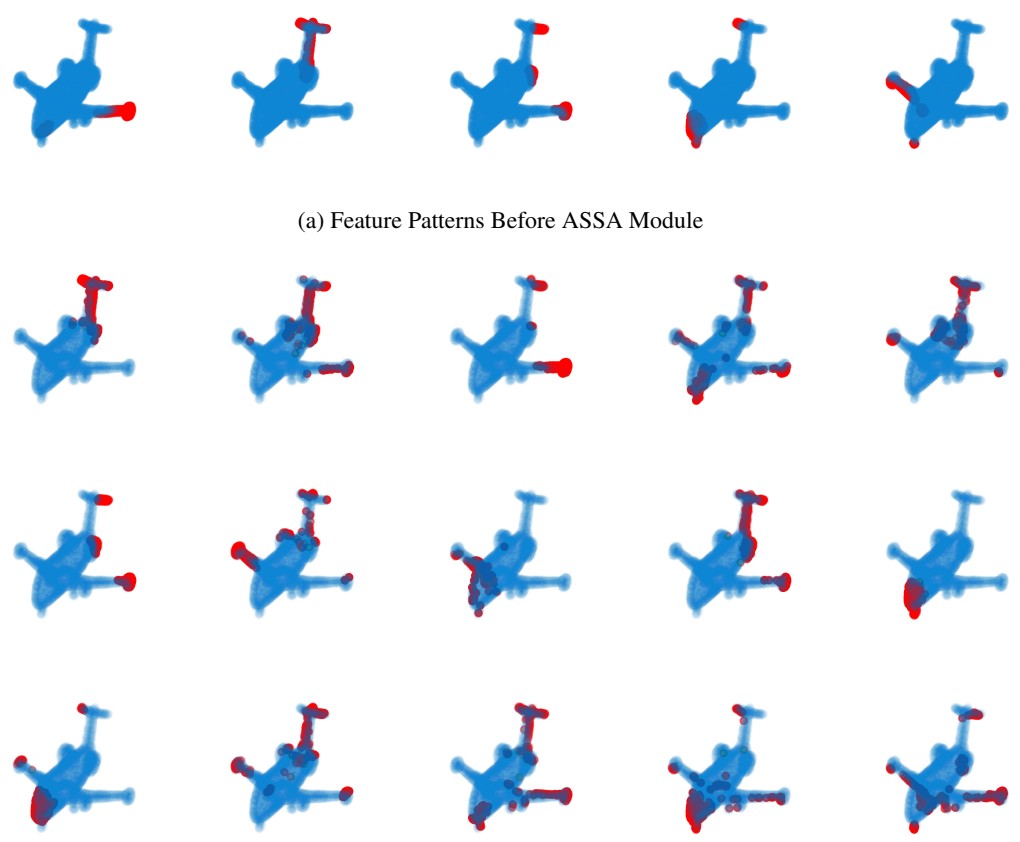

(a) Feature Patterns Before ASSA Module

(b) Feature Patterns After ASSA Module

Figure S3: Airplane feature patterns visualization

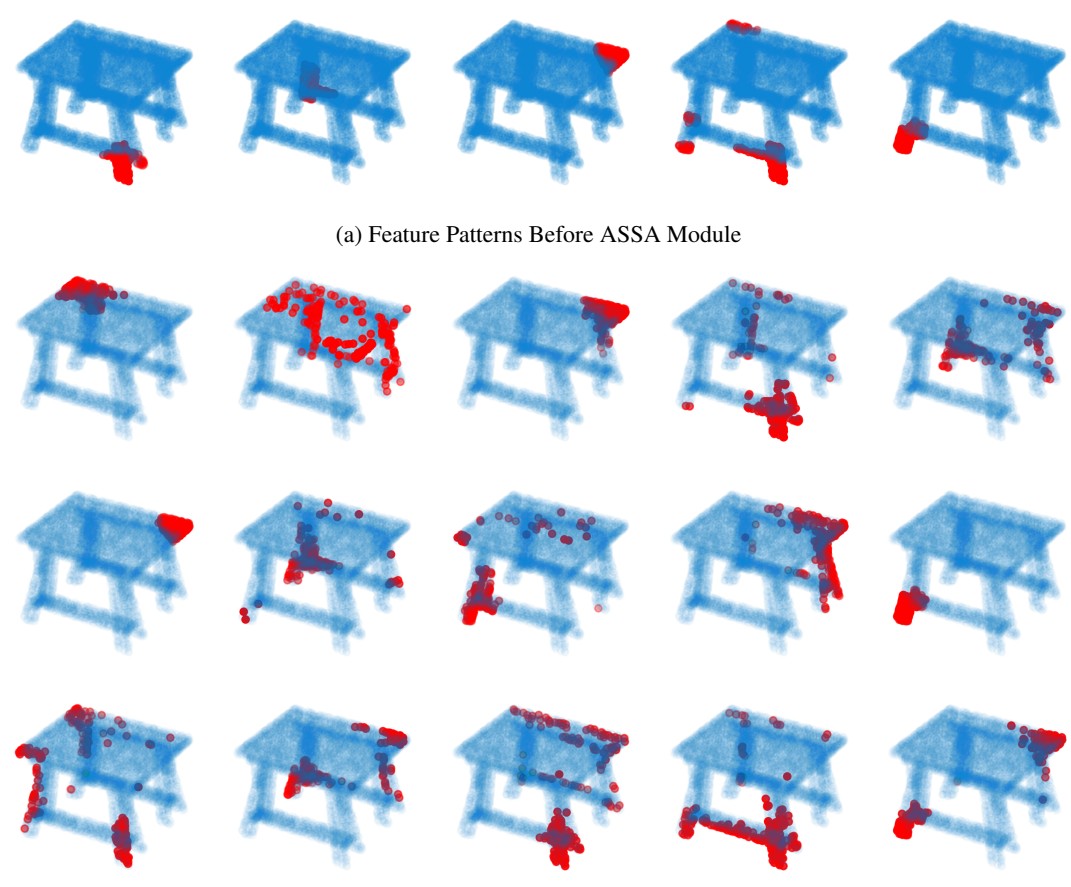

(a) Feature Patterns Before ASSA Module

(b) Feature Patterns After ASSA Module

Figure S4: Table feature patterns visualization

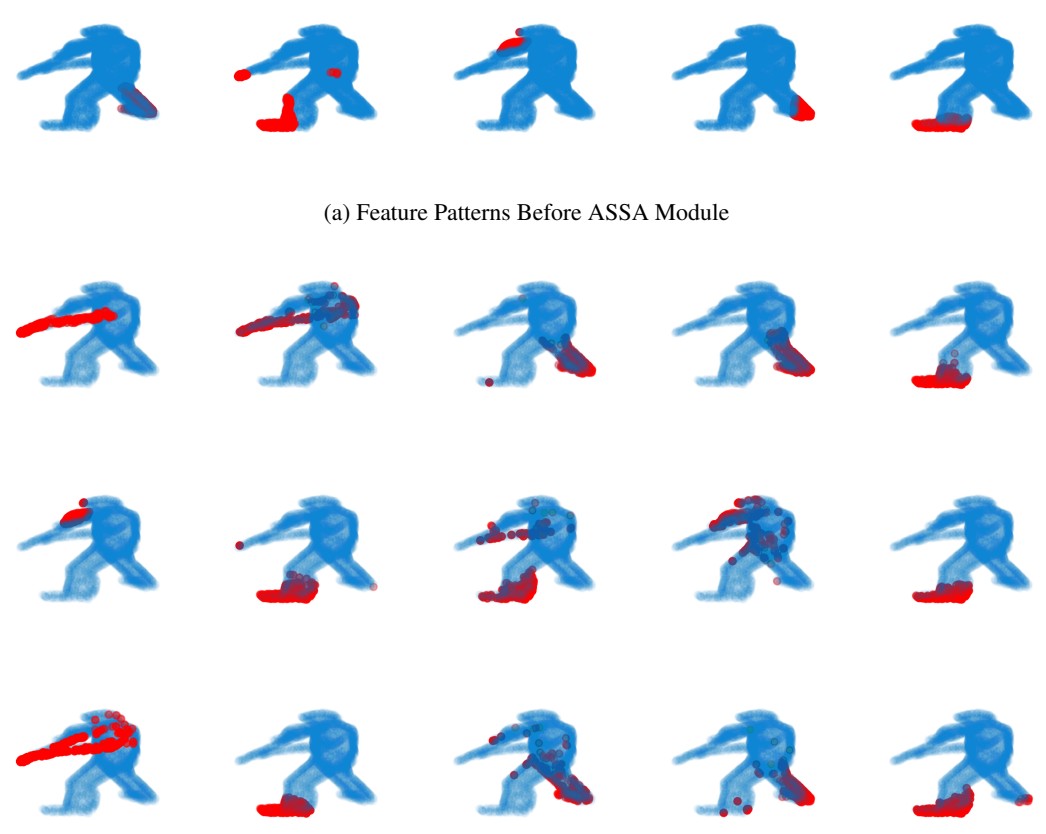

(a) Feature Patterns Before ASSA Module

(b) Feature Patterns After ASSA Module

Figure S5: Human feature patterns visualization

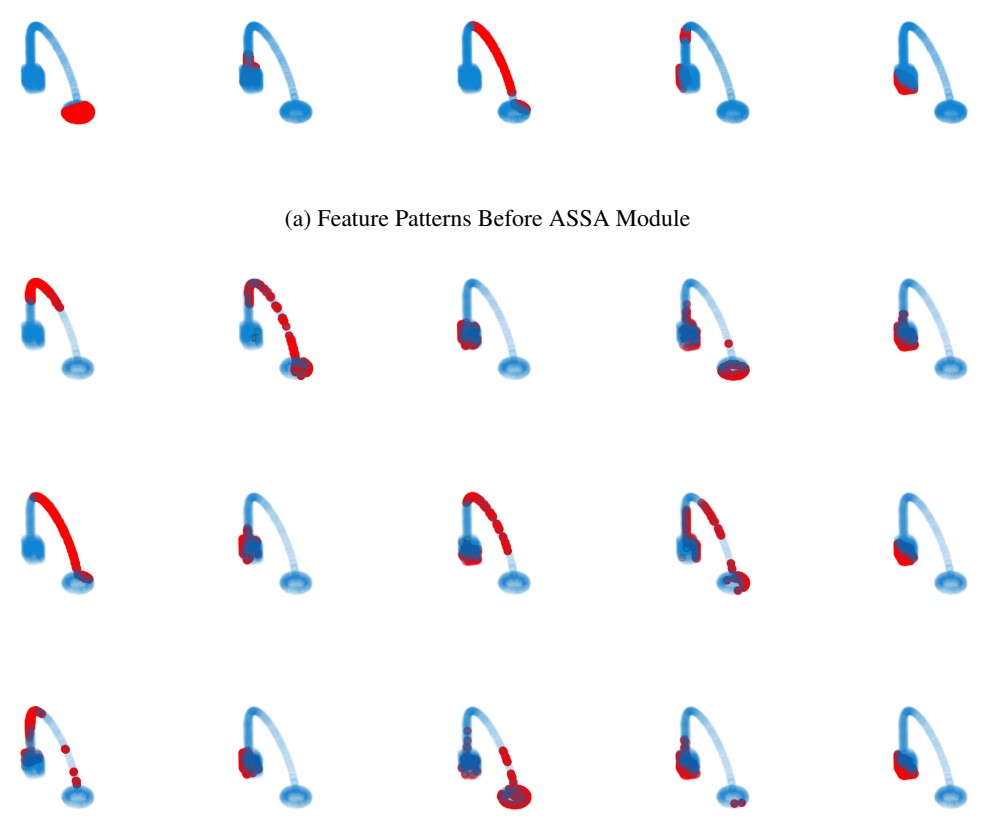

(a) Feature Patterns Before ASSA Module

(b) Feature Patterns After ASSA Module

Figure S6: Lamp feature patterns visualization

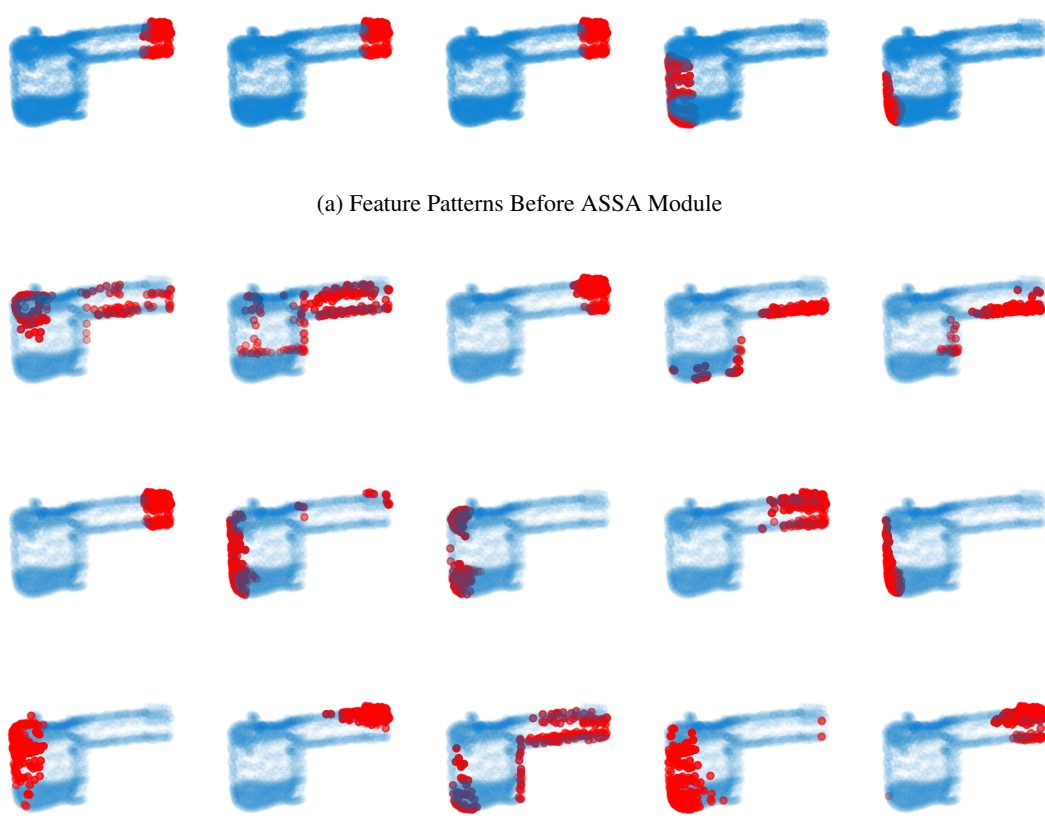

(a) Feature Patterns Before ASSA Module

(b) Feature Patterns After ASSA Module

Figure S7: Guitar feature patterns visualization

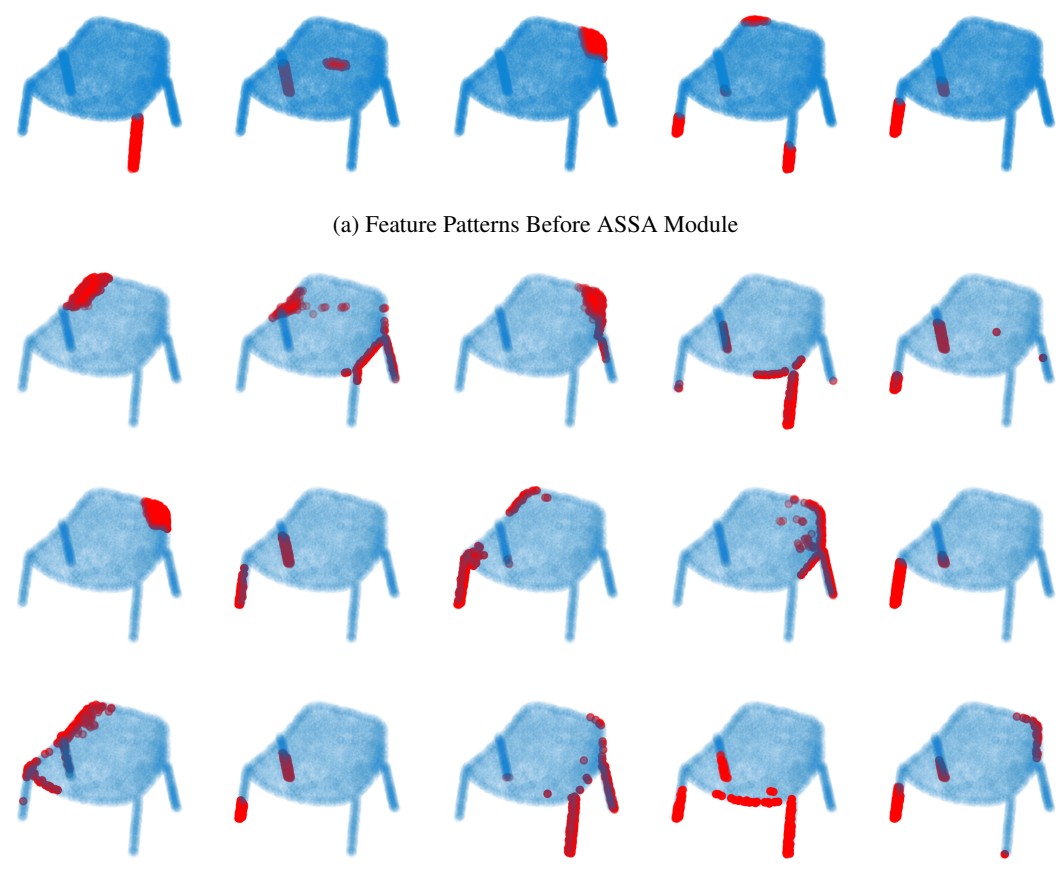

(a) Feature Patterns Before ASSA Module

(b) Feature Patterns After ASSA Module

Figure S8: Chair feature patterns visualization

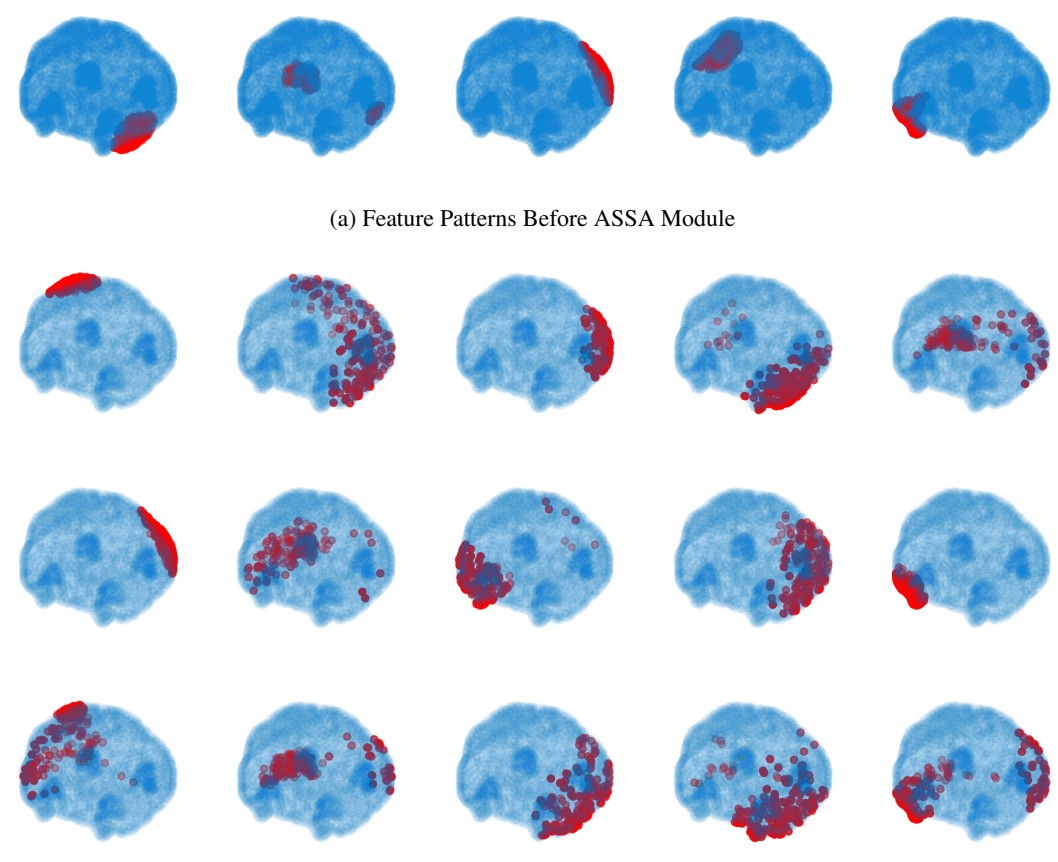

(a) Feature Patterns Before ASSA Module

(b) Feature Patterns After ASSA Module

Figure S9: Car feature patterns visualization