# OpenReview forum: "ASSANet: An Anisotropic Separable Set Abstraction for Efficient Point Cloud Representation Learning"
_NeurIPS.cc/2021/Conference — NeurIPS 2021 Spotlight_

### Official Review · Reviewer_KEKc · 2021-07-09

**Rating:** 7
**Confidence:** 4

**Summary:**

This paper studies the limitations of Set Abstraction in PointNet++. and proposed its variants including PreConv SA, Seperable SA, and Anisotropic Separable SA to reduce the computational cost and improve the accuracy.

**Main Review:**

The proposed method provided an improved version to the original set abstraction and increase the inference speed while maintaining almost the same or even better speed compared to PointNet++ and PosPool.

The paper is well-organized. I think the description of evolution from Vanilla SA and ASSA.

Just a question on Equation (2), repeating f_i^{res} three times seems redundant even though multiplied by the distance in three directions. Have you tried other more efficient ways to incorporate anisotropic property, like using 1/3 channels as in PosPool, or simply addition?


**Time Spent Reviewing:**

5

---

> ### Author Response · Authors · 2021-08-09
> **We tried other possible anisotropic pooling schemes, but they did not work as good as our proposed Anisotropic Reduction**
>
> We thank the reviewer for their insightful comments and time spent reviewing the paper.
>
> Regarding the reviewer's question,  we would like to point out that we have tried multiple schemes before residing on the proposed Anisotropic Reduction presented in the paper. These schemes include:
> 1) PosPool: A comparison between our proposed  Anisotropic Reduction with PosPool is shown in Table 4 (ablation study section - Line 323). Our current scheme achieves higher accuracy with lower latency compared to PosPool.
> 2) We tried out applying simple addition of the relative positions as scaling weights. We obtained 58.5 mIoU on S3DIS Area 5  (compared to 62.6 mIoU using Anisotropic Reduction).
> 3) We also tried using the Euclidean distance between the neighbors and the central point as the scaling weights, however, it achieved around 57.9 mIoU on S3DIS Area 5 (compared to 62.6 mIoU using Anisotropic Reduction).
>
> Furthermore and as suggested by the reviewer, we compared our Anisotropic Reduction with PosPool-like operation using 1/3 channels during the rebuttal period. The mIoU on S3DIS Area 5 is 61.8, which is close to PosPool but worse than our proposed Anisotropic Reduction.
>
> We thank the reviewer for pointing the importance of showing such analysis in the paper and will make these results available in Table 4 of the final version of the paper.

---

> > ### Comment · Reviewer_KEKc · 2021-08-14
> > **Thanks for the response**
> >
> > Thanks the authors for the response. It addressed all my concerns. I am positive towards acceptance of this paper and keep my original rating.

---

### Official Review · Reviewer_sigp · 2021-07-14

**Rating:** 6
**Confidence:** 5

**Summary:**

In this paper, the authors present PointNetV3, which is a faster and more accurate version of PointNet++.  The authors first analyze the latency bottleneck of PointNet++ and find that computation takes up to 70% of total latency, and then propose to reduce the computation cost via several techniques. Namely, separate SA modules are used to reduce the computation cost of vanilla SA (since MLP after reduction is much more efficient); Anisotropic reduction is applied to achieve higher accuracy (original PointNet++ shares conv. weights for all neighbors, which is suboptimal). The authors also analyze the effectiveness of width and depth scaling to improve the efficiency-accuracy tradeoff of PointNetV3 models. Experiment results on S3DIS, ModelNet40 and ShapeNet Part are impressive.

**Limitations And Societal Impact:**

As I have indicated in the main review, the major weakness of this paper is that all experiments are done on small-scale datasets and the number of input points in the point cloud is not large (around 15K). It will be great if the authors can present some clue that the proposed method can scale up to large-scale input, in terms of both efficiency and accuracy.

**Main Review:**

This paper attempts to analyze the efficiency bottleneck of the widely used PointNet++ architecture in 3D deep learning and improve its efficiency-accuracy tradeoff. I feel that the motivation / analysis of this paper is thorough and the proposed solution is simple and effective. The experiment results show that PointNet++-like architecture can achieve significantly better accuracy-efficiency tradeoff comparing with more complicated design such as KPConv, which is interesting. The overall writing quality of this paper is also good.

However, I still believe that there are some challenges to be addressed in order to make this paper more influential.

- The evaluation benchmarks chosen in this paper are all very small datasets, I'm quite curious whether PointNetV3 can be applied to larger datasets such as SemanticKITTI, nuScenes and Waymo to achieve state-of-the-art level accuracy. Even for indoor tasks, I believe ScanNet might be a better benchmark in 2021 comparing with S3DIS.

- It seems that this paper assumes the number of input points per sample is around 15,000. This will be sufficient for most object / indoor-level datasets, but not quite sufficient for LiDAR perception tasks. For example, each scene on SemanticKITTI has ~120K points, and for Waymo each scene contains ~150K points. Can PointNetV3 scale up to these large scale scenes and perform competitively? Notice that at >100K points per scene, neighborhood query might become a significant bottleneck.

- The comparison with point-based methods are really impressive, but SparseConv-based methods are not compared against. These methods are particularly strong at **one-pass inference** for the scenes. That is to say, we do not need to divide the input into sliding windows. I wonder if PointNetV3 can have advantage over SparseConv-based methods in terms of **per-scene latency** (not 16x15000 batched inference latency) and accuracy. Again, it is better to compare at large-scale scenes to make PointNetV3 more influential.

I would raise my score if these concerns (especially the latter two) are addressed properly.

===

I've updated the score to 6 after reading the author response.

**Time Spent Reviewing:**

1.5

---

> ### Author Response · Authors · 2021-08-09
> **PointNetv3 outperforms PointNet++ by over 10 mIoU while being 6 times faster on SemanticKITTI**
>
> We thank the reviewer for their constructive suggestions and comments. We benchmark our proposed PointNetv3 on SemanticKITTI, where it outperforms PointNet++ by a large margin (+10.2 mIoU) while being 6 times faster.
>
> Below we address each comment mentioned by the reviewer in detail:
>
> 1. [Evaluation benchmarks are small datasets]. We are sorry for the inconvenience regarding the choice of the datasets we benchmarked on. We followed what the majority of point-based methods benchmark their methods on. For large-scale datasets, we only benchmarked on S3DIS, which we thought is large enough as it contains around 3 million points for each sub-sampled area. We agree with the reviewer's suggestions that evaluating on the large-scale scenes they suggested can make our work more influential; however, due to limitations in time and computational resources, we chose to test our method on SemanticKITTI, one of the four suggested datasets. The experiments on the other datasets are left as an extension.
>
>
> 2. [Neighborhood query might become a significant bottleneck]. We thank the reviewer for pointing this out. We would note that neighborhood query is not an issue even for  >100K points. This is because ball query is an $\mathcal{O}(N)$ algorithm and can be executed highly in parallel. In Figure 2 of our paper, we showed that the time spent on neighborhood querying does not change much as we increase the number of points. In contrast, point subsampling takes much more time when the number of points increases, since the time complexity of the default farthest point sampling is $\mathcal{O}(N\operatorname{log}N)$ in the worst case scenario. To make point-based methods work efficiently on large-scale scenes, one can simply replace the farthest point sampling in our PointNetv3 with random sampling.  Random sampling has been shown to increase speed while maintaining the accuracy in RandLA-Net (Hu $\textit{et al.}$ , CVPR20).
>
>
> 3. [Comparison with SparseConv-based methods in terms of per-scene latency and accuracy on large-scale datasets]. Using a similar data-processing, training, and evaluation settings as the recent influential SparseConv-based method SPVNAS, we benchmark our proposed PointNetv3 on SemanticKITTI with one-pass inference per scene. Due to limited time, we do not perform any hyper-parameter tuning for our point-based method PointNetv3 and instead directly use the hyper-parameters provided by SPVNAS. As for the architecture, we only modify two parts in PointNetv3: 1) farthest point sampling to random sampling for speed up; 2) the initial radius of ball query from 0.1 meters to 0.5 meters since the scene is larger than S3DIS. With the farthest point sampling, PointNetv3 achieves 29.9 mIoU with 4.2 scenes/second (FPS) on a single GTX1080Ti GPU. With random sampling, PointNetv3 achieves 30.3 mIoU with 6.6 FPS. On the other hand, PointNet++ only achieves 20.2 mIoU and less than 0.1 FPS (sliding-window inference) or 1.1 FPS (one-pass inference).  Our PointNetv3 outperforms PointNet++ by over 10 mIoU while being 6 times faster on SemanticKITTI. Compared to  SparseConv-based methods, SPVNAS (9.1 FPS), PVCNN (6.9 FPS), and MinkowskiNet (3.4 FPS), our PointNetV3 attains comparable speeds.  However, accuracy wise,  PointNetv3 needs some further modifications (tuning hyper-parameters of training and architecture, adding some attention layers to capture the long-range correlations, ...) to reach state-of-the-art level of accuracy on such large-scale datasets.

---

> ### Comment · Reviewer_sigp · 2021-08-12
> **Thanks for the response**
>
> Thanks the authors for the tremendous efforts in the rebuttal. I understand that the rebuttal window is too short for the authors to get good enough results on large-scale datasets. After reading comments from other reviewers and the response, I feel that the merits of this paper outweigh the downsides. As a result, I'll raise the score to 6. I think this paper would have been 8 to 9 if the results on large-scale datasets are comparable / better than SparseConv-based methods.
>
> Small comment for the response: ball query is $O(MN)$ rather than $O(N)$, where $M, N$ are number of points after and before downsampling.

---

### Official Review · Reviewer_APWv · 2021-07-16

**Rating:** 9
**Confidence:** 5

**Summary:**

This paper proposed a novel neural network backbone architecture for 3D point cloud processing that focuses on reducing the repeated computation inside the grouped neighborhood. The author proposed a nice technique to decompose the learning of the relative 3D coordinate values and the learning of features from previous layers so that the majority of repetitive computation can be saved. The experiments are done on several classification and segmentation benchmarks.

**Limitations And Societal Impact:**

No such issue was found.

**Main Review:**

Strong points:

- This paper addressed the problem of repetitive feature computation in PointNet++-style architecture. The problem is important yet has been long ignored. The proposed idea is simple but elegant and effective.

- The experiment results are strong. When viewing performance and model runtime together, the proposed method significantly outperforms previous methods by a notable margin.


Weak points:

- The paper does not include any visualization that helps understand why would the proposed method work. The authors could provide a visualization figure of reduction $\mathcal{R}$ in Equation (2) by showing what area is activated during the reduction operation in a local neighborhood, similar to Figure 19 of the PointNet paper; or by showing the critical point set that is activated, similar to Figure 18 of the PointNet paper.

- It is good that the authors provide Equation (2) for ASSA module, which is the main contribution of the paper. However, the other two proposed PreConv SA and Separable SA are not explained by math equations which makes it really hard for readers to understand and compare with ASSA.

- Figure 3 is very confusing. It does not reflect what ASSA is doing and what difference it is compared to SA and others. I suggest that the authors remove Figure 3 and simply use math equations to explain the four SAs.

- Minor but important notation issue: in Equation (2), the author messed up with the set bracket and function bracket, i.e. the whole set indexed by $j$ should be inside the bracket of $\mathcal{R} ()$ instead of outside of it.

- Minor problem: there should be a line break at line 253?

- Though the author pointed out that the theoretical speed-up of ASSA compared to vanilla SA is $K=32$, why does the actual speed-up is much smaller than this? The author can decompose the actual experiment latency numbers of each part (e.g. data loading, encoder, decoder, etc.) similar to Figure 2 to show the actual speed-up due to ASSA.

- Though this work is on point cloud deep learning, conceptually it is not a direct extension of PointNet and PointNet++. Therefore, it is inappropriate to name this paper "PointNetv3". Instead, the author should emphasize "anisotropic" in the title and the network name. I'm willing to increase the review score to 8 as long as the authors are committed to using another name other than "PointNetv3".

=================================

After rebuttal:

I'm satisfied with the response from the authors. It resolved most of the concerns in my review.

I regard this paper as a very strong submission. It solves an important but long-ignored problem in point cloud deep learning. The solution is beautiful. I'm happy to increase my review score to 9.

There could be further improvement to the paper. Please see my official comment to the authors below.

**Time Spent Reviewing:**

3

---

> ### Author Response · Authors · 2021-08-09
> **Add visualization for understanding, equations for the proposed SA modules, and change the title of our work**
>
> We thank the reviewer for their review and insightful comments. We have addressed the points raised by the reviewer below:
>
> 1. [Visualization for Understanding]. This comment is of special interest to us as it further shows the strength of using our proposed ASSA module and helps understanding why ASSA works.  However, doing something similar to Figures 18 or 19 presented in PointNet in our PointNet++ like architectures is not very insightful. This is mainly attributed to the fact that critical point sets in PointNet++ like methods are simply the sub-sampled coordinates and this is not necessarily useful to understand our ASSA module which improves feature extraction. Instead, we provide feature visualizations similar to Figure 8 (Section 4.4) of PointNet++ paper. The point cloud feature patterns before and after the first ASSA module are shown in [Figure](https://i.ibb.co/yYyjHfn/ASSA-Net-patterns.png) which shows that the ASSA module helps capture better geometric relationships.
>
>
> 2. [Equations of SA variants]. PreConv SA module performs MLPs before the grouping layer and is mathematically represented by:
>   $$
>   \begin{equation}
>   \begin{split}
>   & \mathbf{f}_i^{'} =\operatorname{MLPs}(\mathbf{f}_i^l), \\
>   & \mathbf{f}_i^{l+1} = \mathcal{R}\left(\left\\{\mathbf{f}_j^{'}|j\in \mathcal{N}(i)\right\\}\right),
>   \end{split}
>   \end{equation}
>   $$
> whereas Separable SA evenly separates the MLPs before and after the reduction layer in PreConv SA and then adds a residual connection between the outputs of the two parts of the MLPs and is mathematically represented by:
> $$
> \begin{equation}
> \begin{split}
> & \mathbf{f}_i^{res} =\operatorname{MLPs}(\mathbf{f}_i^l), \\
> & \mathbf{f}_i^{l+1} =\mathbf{f}_i^{res} +
> \operatorname{MLPs}\left(\mathcal{R}\left(\left\\{\mathbf{f}_j^{res}|j\in \mathcal{N}(i)\right\\}\right)\right)
> \end{split}
> \end{equation}
> $$
> The equations will be explicitly added to the paper as suggested by the reviewer.
>
>
> 3. [Figure 3 is confusing]. We are sorry for this. We will revise Figure3 and move it to the appendix.  A more detailed caption will be added.
>
>
> 4. [Equation 2]. Thank you for pointing this out.  We will revise it in the final version.
>
>
> 5. [Line break]. Initially the decision was to remove the line break in order to fit the text into the page limit. This will be fixed in the final version.
>
>
> 6. [Actual speed-up]. We refer the reviewer to Line 230 of our paper. The presented reduction is in terms of FLOPS reduction (factor of 32) and not in terms of speed-up. A 32-factor reduction in FLOPS doesn't necessarily translate to an overall actual speed-up of 32. We note that since the only modified module in our network (compared to PointNet++) is the ASSA module then the speed-up of our model directly reflects the actual speed-up granted by using the proposed module.
>
>
> 7. [Retitle]. We thank the reviewer for the suggestion. The title of our work will be changed to " ASSA-Net: A Fast and Accurate Network for Point Cloud Representation Learning". Instead of referring to our network as PointNetv3, we will now refer to it as ASSA-Net, an Anisotropic Separable Set Abstraction based Network.

---

> > ### Comment · Reviewer_APWv · 2021-08-11
> > **Comment**
> >
> > Thanks to the authors for the reply. I would like to provide more comments for improving the paper.
> >
> > Regarding point 1: Thanks to the authors for providing the visualization. It looks good. The authors should consider include more examples in the paper (or supplementary) besides the provided airplane example. It really helps understand the modules proposed in this paper.
> >
> > Regarding point 6: The author could make a separate table or add a column to the existing table to show the speed-up of ASSA module. I understand that the speed-up of the whole network is not as large as $K=32$ because of other components in the network. The authors can still show the speed-up/latency decomposition of **ASSA module part only** compared to **SA module part only**. There could be a larger speed-up to better reflect the value of ASSA.
> >
> > Regarding point 7: Thanks to the authors for agreeing to change the network name to "ASSA-Net". However, I think "A Fast and Accurate Network for Point Cloud Representation Learning" is still not good: it does not convey any scientific value/contribution of the proposed module (because any published modules are supposed to be fast and accurate). I suggest that the authors include "Anisotropic" and "Separable" in the title to reflect the main contribution, explain the name "ASSA-Net", and distinguish this paper from other publications.

---

> > > ### Author Response · Authors · 2021-08-12
> > > **We highly appreciate the reviewer's comments and address all the concerns**
> > >
> > > We thank the reviewer for their continued constructive comments which are valued as they help make our paper more influential.
> > >
> > > Q1. [More visualization examples]. Below we provide five more visualizations on different classes (car, chair, table, lamp, and person) which will also be included in the supplementary material. The examples can be found at the following links: [car](https://i.ibb.co/YdcksPn/ASSA-Net-patterns-1.png), [chair](https://i.ibb.co/TKz0fNS/ASSA-Net-patterns-2.png), [table](https://i.ibb.co/kDjNBbj/ASSA-Net-patterns-3.png), [lamp](https://i.ibb.co/HNRJdZQ/ASSA-Net-patterns-4.png), and [person](https://i.ibb.co/dJMJgPk/ASSA-Net-patterns-5.png).
> > >
> > >
> > > Q6. [Speed-up decomposition of ASSA module part only]. We thank the reviewer for this great suggestion as it better highlights the speed improvement introduced by our ASSA module. The [link](https://i.ibb.co/z6wzWJL/Latency-Decomposition.png) shows the latency decomposition figure of the ASSA module compared with the SA module.  Two cases are provided: 1) 4096 points as input; 2) 15,000 points as input.  We show the latency of the first ASSA /SA module in the network. In both cases, one can observe a clear reduction of time (around 6 times) in the computation when using the proposed ASSA module. This clearly reflects the efficiency of our anisotropic separable operation.  It is also worth mentioning that since our ASSA module uses the query coordinates as the support coordinates, we also save a small amount of time in grouping (neighborhood querying).
> > >
> > >
> > > Q7. [retitle].  We agree that the explicit use of "Anisotropic" and "Separable" keywords in the title would better reflect the main contribution. Therefore, we rename the paper as: “Anisotropic Separable Set Abstraction for Efficient Point Cloud Representation Learning”. We would appreciate your feedback regarding this new title.

---

### Official Review · Reviewer_ABYL · 2021-07-17

**Rating:** 8
**Confidence:** 3

**Summary:**

The authors modify PointNet++ by replacing the vanilla SA layers in PointNet++ by a new  Anisotropic Separable Set Abstraction layer. The obtained neural network, called PointNetv3, is faster and more accurate.

**Limitations And Societal Impact:**

yes

**Main Review:**

PointNet++ which is a variant of PointNet is a notable neural network using for processing point clouds. PointNet++ contains two main components: Set Abstraction SA (to sub-samples the point clouds and groups the neighborhood) and a local feature extraction (which is PointNet). This paper aims to modify the SA layer to increase the efficiency and accuracy.

The authors propose a novel Anisotropic Separable Set Abstraction (ASSA) component to replace the SA. First, the MLPs in the SA layer are separated to before and after the reduction layer and are connected by a residual connection. Second, the reduction layer is replaced by an Anisotropic Reduction layer to effectively aggregate local information. This ASSA is then used to replace the SA layer in PointNet++ to obtain PointNetv3.


Various experiments with classification on ModelNet40, semantic segmentation on S3DIS and part segmentation on ShapeNet shows that the modification is indeed very useful and PointNetv3 achieves better accuracy and runs faster in comparison with PointNet++ and previous methods.

The paper is very well-written. The result is novel and interesting.

**Time Spent Reviewing:**

8

---

> ### Author Response · Authors · 2021-08-09
> **Thank you for recognizing and appreciating our work.**
>
> We thank the reviewer for their kind review and for recognizing and appreciating the contributions provided by our work.

---

### Decision · Program_Chairs · 2021-09-27

**Decision:**

Accept (Spotlight)

**Comment:**

The paper presents a new variant of PointNet that's faster and more accurate. This paper received positive reviews and all reviewers recommended acceptance. The reviewers find many positive points including impressive empirical results. AC does not find grounds to overturn this consensus recommendation. The authors should incorporate the suggestions of the reviews when revising the paper for the camera ready version.